# Aryl Hydrocarbon Receptor Activation Downregulates IL-33 Expression in Keratinocytes via Ovo-Like 1

**DOI:** 10.3390/jcm9030891

**Published:** 2020-03-24

**Authors:** Gaku Tsuji, Akiko Hashimoto-Hachiya, Vu Hai Yen, Sho Miake, Masaki Takemura, Yasutaka Mitamura, Takamichi Ito, Maho Murata, Masutaka Furue, Takeshi Nakahara

**Affiliations:** 1Department of Dermatology, Graduate School of Medical Sciences, Kyushu University, Fukuoka 812-8582, Japan; yenvuhai@dermatol.med.kyushu-u.ac.jp (V.H.Y.); s-miake@dermatol.med.kyushu-u.ac.jp (S.M.); take0917@dermatol.med.kyushu-u.ac.jp (M.T.); mitamura@dermatol.med.kyushu-u.ac.jp (Y.M.); takamiti@dermatol.med.kyushu-u.ac.jp (T.I.); muratama@dermatol.med.kyushu-u.ac.jp (M.M.); furue@dermatol.med.kyushu-u.ac.jp (M.F.); nakahara@dermatol.med.kyushu-u.ac.jp (T.N.); 2Research and Clinical Center for Yusho and Dioxin, Kyushu University Hospital, Fukuoka 812-8582, Japan; ahachi@dermatol.me.kyushu-u.ac.jp; 3Division of Skin Surface Sensing, Department of Dermatology, Graduate School of Medical Sciences, Kyushu University, Fukuoka 812-8582, Japan

**Keywords:** atopic dermatitis, aryl hydrocarbon receptor, IL-33

## Abstract

Background: IL-33, one of the IL-1 superfamily cytokines, has been shown to be associated with pruritus and inflammation in atopic dermatitis (AD). Furthermore, IL-33 production derived from keratinocytes reportedly has a crucial role in the development of AD; however, the mechanism of IL-33 expression has not been fully understood. Methods: We analyzed IL-33 expression in normal human epidermal keratinocytes (NHEKs) treated with IL-4. Results: IL-4 induced the upregulation of IL-33 expression in NHEKs. Based on the findings 1) that ovo-like 1 (OVOL1), a susceptible gene of AD, upregulates filaggrin (FLG) and loricrin (LOR) expression in NHEKs and 2) that reduced expression of FLG and LOR leads to production of IL-1 superfamily cytokines, we examined the involvement of OVOL1 in IL-33 expression in NHEKs. Knockdown of OVOL1 induced upregulation of IL-33 expression. Moreover, because Glyteer, an activator of aryl hydrocarbon receptor (AHR), reportedly upregulates OVOL1 expression, we examined whether treatment with Glyteer inhibited IL-33 expression in NHEKs. Treatment with Glyteer inhibited IL-4-induced upregulation of IL-33 expression, which was canceled by knockdown of either AHR or OVOL1. Conclusions: Activation of the AHR-OVOL1 axis inhibits IL-4-induced IL-33 expression, which could be beneficial for the treatment of AD.

## 1. Introduction

Atopic dermatitis (AD) is characterized by pruritus, cutaneous inflammation, and dry skin with epidermal barrier dysfunction [1]. Although pruritus is the critical feature of AD that profoundly impairs the quality of life of patients [2], no strategy for controlling it in AD has been established. Recently, IL-33, one of the IL-1 superfamily cytokines, was identified as a pruritus-associated molecule that excites sensory neurons and mediates itch response [3]. IL-33 is stored in the nucleus and can be released after cell necrosis or damage, which may be caused by physical trauma or stress such as scratching of the skin in response to pruritus [4]. It has also been reported that IL-33 expression is increased in the epidermis of patients with AD [5] and that IL-33 production derived from keratinocytes has an important role in the development of AD-like skin lesions in experimental murine models [6,7]. This suggests the benefits of revealing the regulatory mechanism of IL-33 expression in keratinocytes, which could enable the control of pruritus and cutaneous inflammation in AD.

Genome-wide association studies on AD in different ethnic groups have identified genes conferring susceptibility to this disease, including filaggrin (FLG) and OVO-like 1 (OVOL1) [8,9,10]. Previously, we demonstrated that the impairment of OVOL1, a transcription factor profoundly related to epithelial differentiation, resulted in downregulation of the expression of FLG and loricrin (LOR), which are essential for skin barrier formation in AD [11,12,13,14]. Since reduced FLG and LOR expression leads to enhanced expression of IL-1 cytokines such as IL-1alpha and IL-18 in keratinocytes [15], we hypothesized that OVOL1 expression is involved in the expression of IL-33 in keratinocytes in AD.

To demonstrate that, we analyzed IL-33 expression in normal human epidermal keratinocytes (NHEKs) treated with IL-4, an important Th2 cytokine; such treatment can be used to establish an in vitro AD model [11]. We found that the downregulation of OVOL1 expression induced the phosphorylation of extracellular signal-regulated kinase (ERK)-1/2, leading to the upregulation of IL-33 expression. This suggests that OVOL1 is a negative regulator of IL-33 expression induced by IL-4. Furthermore, based on our previous finding that the activation of aryl hydrocarbon receptor (AHR), a ligand-activated transcription factor, upregulates OVOL1 expression [11,12,13], we examined whether the activation of AHR by Glyteer, a soybean tar that is utilized for AD treatment clinically in Japan [16,17], inhibited IL-4-induced IL-33 expression in NHEKs. The results showed that Glyteer treatment inhibited such expression in a manner dependent on AHR and OVOL1 expression. These findings suggest that activation of the AHR-OVOL1 axis exerts an inhibitory effect on IL-33 expression induced by IL-4, which contributes to attenuate pruritus and disease activity in AD.

## 2. Experimental Section

### 2.1. Reagents and Antibodies

Human recombinant IL-4 was purchased from PeproTech (Rocky Hill, NJ, USA). Dimethyl sulfoxide (DMSO) was purchased from Nacalai Tesque (Kyoto, Japan). Tofacitinib (Selleck Chemicals, Houston, TX, USA), JTE-052 (MedChemExpress, Monmouth, NJ, USA), U0126 (Cell Signaling Technology, Danvers, MA, USA), PD184352 (Sigma-Aldrich, St. Louis, MO, USA), SB203580 (Adipogen Lifesciences, San Diego, CA, USA), and SP600125 (Cayman Chemical, Ann Arbor, MI, USA) were dissolved in DMSO and stored at −30 °C until used in the experiments. Glyteer was provided as an original stock solution by Fujinaga Pharm Co., Ltd. (Tokyo, Japan). Since Glyteer is dry distillation tar of delipidated soybean, it is thought to consist of a wide range of organic compounds and polycyclic aromatic hydrocarbons. However, no further attempts have been made to quantify the compound fraction of Glyteer. Glyteer was administered to culture medium directly to reach final concentrations of 10^−5^%, 10^−6^%, and 10^−7^%. The percentage here indicates the volume % of Glyteer in the culture medium. Anti-human IL-33 monoclonal mouse antibody (Abcam, Cambridge, UK) and anti-human CCL26 rabbit polyclonal antibody (MyBioSource, San Diego, CA, USA) were used for immunofluorescence staining. Anti-human IL-33 monoclonal mouse antibody (Abcam), anti-phosphorylated ERK-1/2 rabbit monoclonal antibody (Thr202/Tyr204) (Cell Signaling Technology), anti-ERK 1/2 rabbit monoclonal antibody (Cell Signaling Technology), anti-phosphorylated p38 rabbit monoclonal antibody (Thr180/Tyr182) (Cell Signaling Technology), anti-p38 rabbit monoclonal antibody (Cell Signaling Technology), anti-phosphorylated c-Jun N-terminal Kinase (JNK) rabbit monoclonal antibody (Thr183/Tyr185) (Cell Signaling Technology), anti-JNK rabbit monoclonal antibody (Cell Signaling Technology), and anti-β-actin monoclonal mouse antibody (Cell Signaling Technology) were used for Western blotting.

### 2.2. Cell Culture

Normal human epidermal keratinocytes (NHEKs) obtained from Lonza (Basel, Switzerland). NHEKs were cultured in serum-free keratinocyte growth medium (Lonza) supplemented with bovine pituitary extract, recombinant epidermal growth factor, insulin, hydrocortisone, transferrin, and epinephrine. Culture medium was replaced every 2–3 days. Cells approaching confluence (70–90%) were disaggregated with 0.25 mg/mL trypsin/0.01% ethylenediaminetetraacetic acid and subcultured. Second- to fourth-passage NHEKs were used in all experiments. NHEKs were seeded, allowed to attach for 24 h, and then utilized for further experiments.

### 2.3. Cell Viability Assay

To evaluate cell viability, we utilized the WST-1 cell proliferation assay system (Takara Bio, Shiga, Japan) in accordance with the manufacturer’s protocol. Cell viability is described in Appendix A.

### 2.4. Immunofluorescence and Confocal Laser Scanning Microscopy

NHEKs were cultured on slides (Lab-Tek, Rochester, NY, USA). These slides were then washed in phosphate-buffered saline (PBS), fixed with acetone for 10 min, and blocked using 10% bovine serum albumin (Roche Diagnostics, Basel, Switzerland) in PBS for 30 min for staining of CCL26. For IL-33 staining, the slides were fixed with 4% paraformaldehyde and then permeabilized using 0.1% saponin (Nacalai tesque) for 6 min. Samples were incubated with either primary anti-human CCL26 rabbit polyclonal antibody (1:100) (MyBioSource) or primary anti-human IL-33 mouse monoclonal antibody (1:100) (Abcam) in WesternBreeze Blocker/Diluent (Invitrogen, Carlsbad, CA, USA) overnight at 4 °C. The slides were then washed with PBS before incubation with anti-rabbit secondary antibody (Alexa Fluor 546; Molecular Probes, Eugene, OR, USA) for 1 h at room temperature. After nuclear staining with 4′,6-diamidino-2-phenylindole (DAPI), the slides were mounted with UltraCruz mounting medium (Santa Cruz Biotechnology, Dallas, TX, USA). All samples were analyzed using a D-Eclipse confocal laser scanning microscope (Nikon, Tokyo, Japan).

### 2.5. qRT-PCR

Total RNA was extracted using the RNeasy Mini kit (Qiagen, Hilden, Germany). Reverse transcription was performed using PrimeScript RT-reagent kit (Takara Bio). qRT-PCR was performed on the CFX connect real-time system (Bio-Rad, Hercules, CA, USA) using TB Green Premix Ex Taq (Takara Bio). Amplification was started at 95 °C for 30 s as the first step, followed by 40 cycles of qRT-PCR at 95 °C for 5 s and 60 °C for 20 s. mRNA expression was measured in triplicate and was normalized to β-actin expression levels. The primer sequences are shown in Appendix A.

### 2.6. Transfection with siRNAs against AHR and OVOL1

Small interfering RNAs (siRNAs) against AHR (si-AHR, s1200), OVOL1 (si-OVOL1, s9939), as well as siRNA consisting of a scrambled sequence that would not lead to specific degradation of any cellular message (control siRNA (si-control)) were purchased from Ambion (Austin, TX, USA). NHEKs were incubated with a mixture of HiPerFect Transfection reagent (Qiagen) culture medium. After incubation for 48 h, siRNA-transfected NHEKs were utilized for further experiments. The knockdown efficiency of si-AHR and si-OVOL1 was shown in Appendix A, which is consistent with our previously report [11].

### 2.7. Western Blotting Analysis

NHEKs were incubated for 5 min in lysis buffer (Complete Lysis M; Roche Diagnostics, Basel, Switzerland). The lysate protein concentration was measured with a BCA protein assay kit (Thermo Fisher Scientific, Waltham, MA, USA). Equal amounts of protein were dissolved in NuPage LDS sample buffer (Invitrogen) and 10% NuPage sample reducing agent (Thermo Fisher Scientific). Lysates were boiled at 70 °C for 10 min and loaded and run on Bis-Tris Gel (Thermo Fisher Scientific) at 200 V for 20 min. The proteins were transferred to PVDF membrane (Merck Millipore, Burlington, MA, USA) and blocked in Western breeze blocker/diluent (Invitrogen). Membranes were probed with anti-human IL-33 mouse monoclonal antibody (Abcam), anti-phosphorylated ERK-1/2 rabbit monoclonal antibody (Thr202/Tyr204) (Cell Signaling Technology), anti-ERK 1/2 rabbit monoclonal antibody (Cell Signaling Technology), anti-phosphorylated p38 rabbit monoclonal antibody (Thr180/Tyr182) (Cell Signaling Technology), anti-p38 rabbit antibody(Cell Signaling Technology), anti-phosphorylated rabbit JNK antibody (Thr183/Tyr185) (Cell Signaling Technology), anti-JNK rabbit monoclonal antibody (Cell Signaling Technology), anti-human AHR rabbit monoclonal antibody (Cell Signaling Technology), anti-human OVOL1 mouse monoclonal antibody (Abcam), or anti-human β-actin mouse monoclonal antibody (Cell Signaling Technology) overnight at 4 °C. Anti-mouse and anti-rabbit horseradish peroxidase-conjugated IgG antibodies (Cell Signaling Technology) were used as secondary antibodies. Visualization of protein bands was accomplished with Chemi-Lumi One Super (Nacalai Tesque) using the ChemiDoc Touch Imaging System (Bio-Rad).

### 2.8. Statistical Analysis

Unpaired Student’s t-test (when two groups were analyzed) and one-way ANOVA (for three or more groups) were used to analyze the results, with a *p*-value less than 0.05 being considered to indicate a statistically significant difference.

## 3. Results

### 3.1. IL-4 Induces Upregulation of IL-33 Expression in NHEKs

To investigate the regulatory mechanism of IL-33 expression induced by IL-4, a key cytokine in the pathogenesis of AD [1], we examined this expression in NHEKs treated with IL-4. NHEKs were treated with PBS (control) or IL-4 (10 ng/mL) for 6, 8, 16, 24, 32, and 48 h for quantitative real-time PCR (qRT-PCR) and for 24 and 48 h for Western blotting analysis. IL-4 induced the upregulation of IL-33 mRNA (Figure 1A) and protein (Figure 1B). Furthermore, NHEKs were treated with IL-4 at different doses (0.1, 1, and 10 ng/mL) for 24 h. IL-4 induced the upregulation of IL-33 mRNA in a dose-dependent manner (Figure 1C). Western blotting analysis confirmed that IL-4 induced the upregulation of IL-33 protein in NHEKs treated with IL-4 (0.1, 1, and 10 ng/mL) for 24 h (Figure 1D). Confocal laser scanning microscopy analysis revealed that IL-33 was not expressed in the nucleus under unstimulated conditions (Figure 1F). Following treatment with IL-4 (10 ng/mL) for 8 h (Figure 1G) and 24 h (Figure 1H), enhanced IL-33 expression was observed in the nucleus. These findings are consistent with previous reports regarding IL-4-induced upregulation of IL-33 expression in NHEKs [18,19].

### 3.2. Upregulation of IL-33 Expression by IL-4 Is Unlikely to Require JAK/STAT6 Axis Activation in NHEKs

Since IL-4 activates the JAK/STAT6 axis upon binding to IL-4 receptor alpha [20], we examined whether IL-4 induced the upregulation of IL-33 expression via the JAK/STAT6 axis. For this, NHEKs were treated with IL-4 (10 ng/mL) for 24 h in the presence or absence of tofacitinib (100, 300, and 500 nM) or JTE-052 (100, 500, and 1000 nM), which are JAK inhibitors clinically utilized for the treatment of inflammatory diseases [21,22,23], for qRT-PCR and Western blotting analyses. Treatment with either tofacitinib or JTE-052 did not affect but rather upregulated IL-33 mRNA (Figure 2A,D) and protein (Figure 2B,E) levels induced by IL-4. However, treatment with either tofacitinib or JTE-052 successfully suppressed IL-4-induced upregulation of the mRNA of CCL26 (Figure 2C,F), a representative JAK/STAT6 axis-mediated chemokine [24,25] reflecting disease activity in AD [26], in a dose-dependent manner. The production of CCL26 was also evaluated by immunofluorescence staining using anti-CCL26 antibody. Treatment with either tofacitinib or JTE-052 inhibited the production of CCL26 induced by IL-4 (Appendix A). These results imply that the upregulation of IL-33 expression by IL-4 is unlikely to require JAK/STAT6 axis activation in NHEKs.

### 3.3. Upregulation of IL-33 Expression by IL-4 Is Dependent on ERK-1/2 and p38 Signaling Pathway in NHEKs

It has been reported that IL-4 also activates MAPKs including ERK1/2, p38, and JNK signaling pathway in NHEKs [27]. Next, we examined phosphorylation of ERK-1/2, p38, and JNK induced by IL-4 stimulation in NHEKs. NHEKs were treated with PBS (control) or IL-4 (10 ng/mL) for 10, 20, 30, and 60 min for Western blotting of phosphorylated ERK-1/2, p38, and JNK. IL-4 induced phosphorylation of ERK-1/2, p38, and JNK in a time-dependent manner (Figure 3A–C).

To further examine whether IL-4 regulates IL-33 expression via ERK-1/2, p38, and JNK signaling pathway, we utilized U0126, a specific inhibitor of MEK that acts upstream of ERK-1/2, SB203580, a specific inhibitor of p38, and SP600125, a specific inhibitor for JNK. NHEKs were treated with IL-4 (10 ng/mL) for 24 h in the presence or absence of U0126 (10 μM), SB203580 (10 μM), and SP600125 (1 μM) for qRT-PCR analysis. Treatment with either U0126 or SB203580 inhibited upregulation of IL-33 mRNA (Figure 3D,F) and protein (Figure 3E,G) induced by IL-4; in contrast, treatment with SP600125 did not inhibit upregulation of IL-33 mRNA (Figure 3H) and protein (Figure 3I) induced by IL-4. PD184352, another MEK inhibitor, also inhibited upregulation of IL-33 mRNA (Appendix A) and protein (Appendix A) induced by IL-4. These results suggest that upregulation of IL-33 expression induced by IL-4 is dependent on activation of ERK-1/2 and p38 but not the JNK signaling pathway in NHEKs.

### 3.4. Downregulation of OVOL1 Expression Leads to Enhanced Upregulation of IL-33 Expression Induced by IL-4

Based on previous reports including ours describing (1) that OVOL1 positively regulates the expression of FLG and LOR in NHEKs [11,12,13,14] and (2) that reduced expression of FLG and LOR is associated with the production of IL-1 superfamily cytokines such as IL-1alpha and IL-18 in keratinocytes [15], we speculated that OVOL1 may be involved in the expression of IL-33, which is also an IL-1 superfamily cytokine. To examine this, we knocked down OVOL1 by siRNA transfection. After transfection with either siRNA control (si-control) or siRNA against OVOL1 (si-OVOL1) for 48 h, NHEKs were treated with IL-4 (10 ng/mL) for 24 h for qRT-PCR and Western blotting analyses. IL-33 mRNA and protein levels were increased in si-OVOL1-transfected NHEKs compared with the levels in si-control-transfected NHEKs (Figure 4A,B). Moreover, the upregulation of IL-33 mRNA and protein induced by IL-4 was enhanced in si-OVOL1-transfected NHEKs compared with that in si-control-transfected NHEKs (Figure 4A,B). Since there is a possibility that knockdown of OVOL1 may modify activation of IL-4 receptor alpha (IL-4RA)/STAT6 signaling pathway induced by IL-4, we evaluated expression of IL-4RA and STAT6 mRNA in si-OVOL1-transfected NHEKs; however, the knockdown of OVOL1 did not alter them (Appendix A).

To further examine whether OVOL1 affects the phosphorylation of ERK-1/2, p38, and JNK induced by IL-4 leading to the upregulation of IL-33 expression, NHEKs transfected with either si-control or si-OVOL1 were treated with IL-4 (10 ng/mL) for 10, 20, 30, or 60 min for Western blotting analysis. The results showed that the phosphorylation of ERK-1/2 was enhanced in si-OVOL1-transfected NHEKs compared with that in si-control-transfected NHEKs (Figure 4C). Furthermore, IL-4 induced the phosphorylation of ERK-1/2 at 10 min, which was gradually diminished within 30 min in si-control-transfected NHEKs; in contrast, the IL-4-induced phosphorylation of ERK-1/2 was detected for more than 30 min in si-OVOL1-transfected NHEKs (Figure 4C). In contrast, phosphorylation of p38 and JNK induced by IL-4 were not affected in si-OVOL1-transfected NHEKs compared with that in si-control-transfected NHEKs (Figure 4D,E). These data indicate that sustained activation of ERK-1/2 by OVOL1 downregulation contributes to enhanced expression of IL-33 in si-OVOL1-transfected NHEKs. To further determine whether ERK-1/2 activation is involved in the mechanism, we examined whether treatment of U0126 and PD184352 may affect the enhanced upregulation of IL-33 expression induced by IL-4 in OVOL1-knockdown NHEKs. NHEKs were transfected with si-control, and si-OVOL1 were treated with IL-4 (10 ng/mL) for 24 h in the presence or absence of either U0126 (10 μM) or PD184352 (10 μM) for qRT-PCR and Western blotting analyses of IL-33. Treatment of U0126 inhibited IL-4-induced upregulation of IL-33 mRNA (Figure 4F) and protein (Figure 4G) levels in si-control- and si-OVOL1-transfected NHEKs. Similar results are obtained from treatment of PD184352 (Appendix A). These results suggest that the downregulation of OVOL1 enables sustained activation of the ERK-1/2 signaling pathway, leading to the enhanced upregulation of IL-33 expression in NHEKs.

### 3.5. Glyteer, an Activator of the AHR-OVOL1-FLG Axis, Inhibits the Upregulation of IL-33 Expression Induced by IL-4 in NHEKs

Glyteer is a soybean tar clinically utilized for the treatment of AD [16,17]. Our previous study showed that Glyteer upregulates the mRNA and protein levels of OVOL1 via aryl hydrocarbon receptor (AHR) activation in NHEKs [11]. Therefore, we tested whether treatment with Glyteer influences IL-33 expression in NHEKs. NHEKs were treated with IL-4 (10 ng/mL) for 24 h in the presence or absence of Glyteer (10−^7^%, 10^−6^%, and 10^−5^%) for qRT-PCR and Western blotting analyses of IL-33. Treatment with Glyteer inhibited the upregulation of the mRNA and protein expression of IL-33 induced by IL-4 (Figure 5A,B).

To further examine whether the inhibitory effect of Glyteer on IL-4-induced upregulation of IL-33 expression is dependent on the AHR-OVOL1 axis, si-control-, si-AHR-, or si-OVOL1-transfected NHEKs were treated with IL-4 (10 ng/mL) for 24 h in the presence or absence of Glyteer (10^−5^%) for qRT-PCR and Western blotting analyses of IL-33. Knockdown of either AHR or OVOL1 canceled this inhibitory effect on IL-4-induced upregulation of IL-33 mRNA (Figure 5C) and protein in NHEKs (Figure 5D,E).

## 4. Discussion

Several studies on the mechanism of IL-33 expression in human keratinocytes have reported that Th1 and Th17 cytokines such as interferon gamma (IFN-gamma) and IL-17A induce the upregulation of IL-33 expression in NHEKs [18,19,28]. This suggests that IL-33 may be involved in the pathogenesis of inflammatory skin diseases, including psoriasis. IL-33 has two functions. One involves its action as an alarmin (alarming cytokine) in the body by extracellular production caused by cell damage such as injury and infection; this shifts immune cells to a Th2-polarized immune response [29]. The other involves it working as a transcription factor repressing inflammatory reactions such as impairment of activation of the nuclear factor-kappa B signaling pathway, leading to the production of proinflammatory cytokines [30]. IL-33 is considered to be one of the critical cytokines causing the development of AD [6,7]. Despite focus being placed on the emerging role of IL-33 in AD, how IL-33 expression is regulated in AD has not been fully clarified.

Several studies have shown that IL-4 induces the upregulation of IL-33 expression in human keratinocytes [18,19]. In addition, in a study utilizing RNA-seq to profile the transcriptome of NHEKs stimulated by various cytokines, significant upregulation of IL-33 expression was observed in those treated with IL-36γ, IL-13, and IL-4 plus IL-13 [31]. Building on these previous studies, here, we confirmed that treatment with IL-4 can induce the upregulation of IL-33 expression in NHEKs (Figure 1). This suggests the existence of a positive feedback loop connecting IL-4 and IL-33 in AD, since IL-33 prompts Th2 lymphocytes, innate lymphoid cells 2, mast cells, and eosinophils to produce Th2 cytokines including IL-4 [29]. Furthermore, IL-4 has recently been determined to stimulate itch-sensory neurons via direct binding to the IL-4 receptor on nerves [32]. Because IL-33 can also stimulate the neurons related to pruritus via the receptor for IL-33 (ST2 receptor) on the nerves [3], there is a possibility that the positive feedback loop of IL-4 and IL-33 may contribute to intense pruritus in AD. Recently, it has been reported that treatment with dupilumab, an anti-IL-4 receptor alpha antibody, induces rapid improvement of itch in AD patients [33]. These lines of clinical evidence support the possibility that the itch in AD may be improved by blockade of the positive feedback loop of IL-4 and IL-33.

Since IL-4 is a potent activator of the JAK/STAT6 axis18, we examined whether either tofacitinib or JTE-052 inhibited IL-4-induced upregulation of IL-33 expression in NHEKs. Although treatment with either tofacitinib or JTE-052 suppressed IL-4-induced upregulation of CCL26 mRNA (Figure 2C) and protein (Appendix A), it did not affect or rather increased IL-4-induced upregulation of IL-33 expression in NHEKs (Figure 2A,B,D,E). This suggests that treatment using JAK inhibitors alone may be insufficient to inhibit IL-33 expression induced by IL-4 in AD. In addition to activation of the JAK/STAT6 axis, it has been reported that IL-4 activates the MAPK signaling pathway including p38 [34], ERK-1/2 [34], and JNK [35]. As shown Figure 3 and Appendix A, the ERK1/2 and p38 signaling pathways were involved in IL-4-induced IL-33 expression, which is consistent with the evidence that IL-17A induces IL-33 expression via the ERK1/2 and p38 signaling pathways in NHEKs [26].

In the present study, we have shown that the downregulation of OVOL1 enabled sustained activation of the ERK-1/2 signaling pathway (Figure 4C). Activation of ERK-1/2 signaling induced by IL-4 has been shown to have a more critical effect on the downregulation of keratin 1 (KRT1), one of the major keratinocyte structural components, compared with p38 and JNK in NHEKs [25]. It has been reported that IL-33 expression is enhanced in the epidermis of Krt1 knockout mouse, suggesting that Krt1 inhibits IL-33 expression in the murine epidermis [36]. Therefore, IL-4-induced activation of the ERK-1/2 signaling pathway and subsequent downregulation of KRT1 may be involved in IL-33 expression in NHEKs.

It has already been reported that reduction of either FLG or LOR expression increases the production of IL-1alpha and IL-18, IL-1 family cytokines, in NHEKs [15]; however, the molecular mechanism behind this has not been determined. Our data suggest that the sustained activation of the ERK-1/2 signaling pathway resulting from the downregulation of OVOL1, a positive regulator of FLG and LOR, is an important clue to understanding the underlying mechanism. However, the mechanism by which OVOL1 negatively mediates activation of the ERK-1/2 signaling pathway has yet to be revealed. Since OVOL1 is a transcription factor, we assumed that it may negatively regulate factors inhibiting ERK phosphorylation such as MAPK scaffolding proteins [37]; however, further study will be needed to confirm this.

Finally, we examined the possibility that the activation of AHR that upregulates OVOL1 expression may control IL-33 expression induced by IL-4 in NHEKs. Several studies including our own have reported that the activation of AHR using AHR ligands attenuates the development of AD in vitro and in vivo [11,37,38,39]. However, whether the activation of AHR inhibits IL-33 expression in AD has not been fully understood. Glyteer, a tar derived from soybean, is utilized for the treatment of AD clinically in Japan [16,17]. We previously reported that the activation of AHR induced by treatment with Glyteer upregulates OVOL1 expression in NHEKs [11,16]. In the present study, we have shown that treatment with Glyteer inhibits IL-4-induced upregulation of IL-33 expression via the AHR-OVOL1 axis in NHEKs (Figure 5). Since the effect of knockdown of AHR on enhanced IL-33 mRNA and protein levels was larger than that of knockdown of OVOL1 (Figure 5C,D), there two possibilities: (1) that knockdown of AHR may modulate IL-4RA/STAT6 signaling pathway and (2) that knockdown of AHR may enhance activation of MAPK signaling pathway in NHEKs. However, the knockdown of AHR did not alter expression of IL-4 RA and STAT6 mRNA (Appendix A) and phosphorylation of ERK1/2, p38, and JNK induced by IL-4 (Appendix A). Therefore, AHR is likely to mediate IL-33 expression transcriptionally in NHEKs; however, further investigations will be needed to demonstrate this.

It has been shown that the induction of IL-33 is mediated via AHR-binding sites located in the IL-33 promoter region. Furthermore, 2,3,7,8-tetrachlorodibenzodioxin (TCDD), diesel exhaust particles (DEPs), and benzo(a)pyrene (BaP) have been shown to induce the upregulation of IL-33 expression [40,41,42]. In contrast, it has been reported that treatment with 6-formylindolo[3,2-b]carbazole (FICZ), an endogenous AHR ligand, does not upregulate IL-33 expression but instead downregulates it in differentiated human keratinocytes, compared with the case in undifferentiated human keratinocytes [38]. Therefore, whether AHR activation results in the inhibition of IL-33 expression is still controversial; however, it has been shown that the type of AHR ligand is an important factor determining the response of AHR signaling [43]. We also examined whether tapinarof, a potent AHR activator [44] that is utilized clinically in the treatment of psoriasis and atopic dermatitis [45,46], has the same effect as Glyteer on IL-4-induced upregulation of IL-33 expression. As shown in Appendix A, tapinarof inhibited the IL-4-induced upregulation of IL-33 expression in NHEKs.

It has been shown that the response of AHR signaling is modified by oxidative stress [41]. TCDD, DEPs, and BaP generate excessive amounts of reactive oxygen species (ROS) and subsequently induce oxidative damage in the cell [47,48,49], since their metabolism by CYP1A1 is unsuccessful because they are structurally stable. Ligation of AHR also activates nuclear factor erythroid 2-related factor 2 (NRF2) and upregulates the expression of antioxidative enzymes [41]. TCDD, DEPs, and BaP activate the AHR-NRF2 battery, but the AHR-CYP1A1 activation may cause far more oxidative stress that cannot be extinguished by the AHR-NRF2 battery. In contrast, many phytochemical AHR ligands stimulate the AHR-NRF2 battery more strongly than AHR-CYP1A1-mediated ROS production and exerts antioxidative effects [44]. Therefore, we believe that oxidative stress induced by AHR activation has an important role in the mechanism of AHR-mediated IL-33 expression. Considering that oxidative stress serves as a key checkpoint for IL-33 production, which is inhibited by NRF2 activation [50], there is a possibility that AHR-NRF2 activation by Glyteer may also be involved in the inhibitory effect on the upregulation of IL-33 expression.

This finding is important to understand a new mechanism by which the activation of AHR attenuates pruritus and disease activity in AD. Taking these findings together, a potent activator of the AHR-OVOL1 axis should be appropriate as a therapeutic agent for the treatment of pruritus and disease activity in AD.

## Figures and Tables

**Figure 1 jcm-09-00891-f001:**
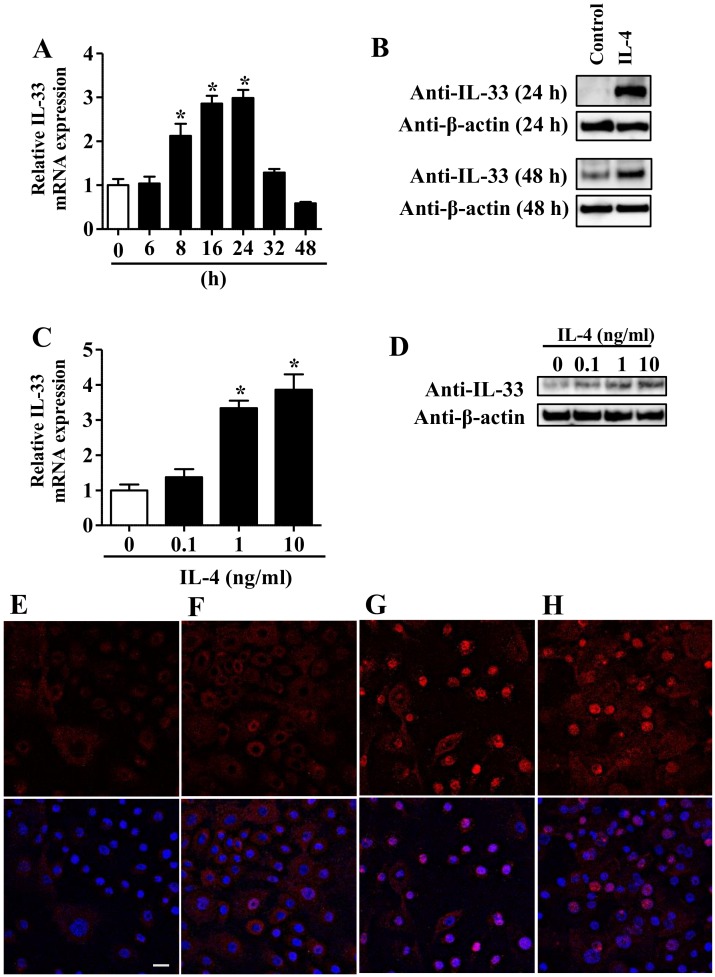
(**A**,**B**) Normal human epidermal keratinocytes (NHEKs) were treated with IL-4 (10 ng/mL) for the indicated period. (**C**,**D**) NHEKs were treated with IL-4 at the indicated dose for 24 h. (**A**,**C**) IL-33 expression was analyzed by qRT-PCR. Data are expressed as mean ± S.E.M.; n = 3 for each group. Statistically significant differences between the expression of control and IL-4-treated NHEKs are presented: * *p* < 0.05. (**B**,**D**) Total cell lysates were prepared and subjected to Western blotting analysis with an anti-IL-33 antibody. The data are representative of experiments repeated three times with similar results. (**E**) Isotype negative control: The scale bar represents 25 μm. NHEKs not treated with IL-4 (**F**), treated with IL-4 (10 ng/mL) for 8 h (**G**), or treated with IL-4 (10 ng/mL) for 24 h (**H**) were stained with an anti-IL-33 antibody (primary antibody) and an Alexa Fluor 546-conjugated anti-mouse IgG antibody (red: secondary antibody). DAPI (4′,6-diamidino-2-phenylindole) was utilized for nuclear staining. Confocal laser scanning images revealed increased nuclear IL-33 expression (red) in IL-4-treated NHEKs compared with that in phosphate-buffered saline (PBS)-treated NHEKs. The data are representative of experiments repeated three times with similar results.

**Figure 2 jcm-09-00891-f002:**
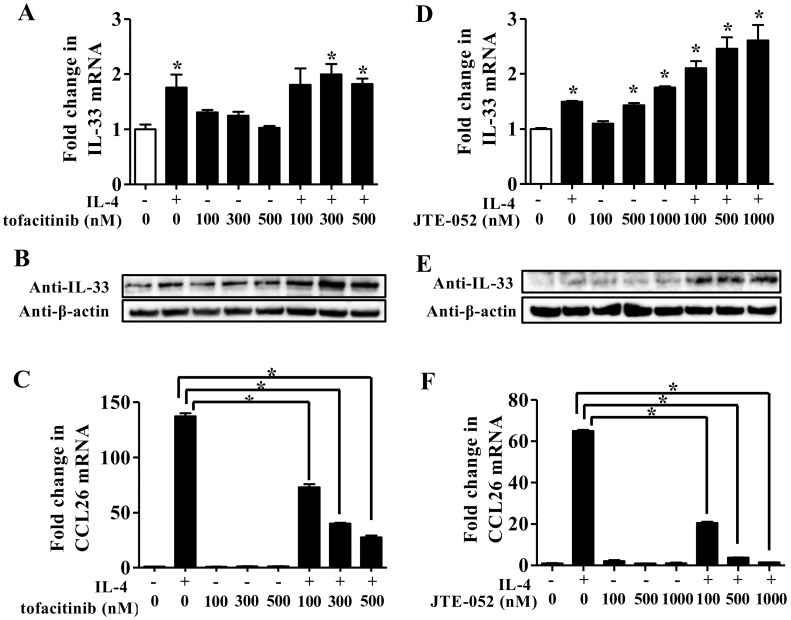
(**A**–**F**) NHEKs were treated with IL-4 (10 ng/mL) in the presence or absence of either tofacitinib (100, 300, and 500 nM) or JTE-052 (100, 500, and 1000 nM) for 24 h. (**A**,**D**) IL-33 mRNA expression was analyzed by qRT-PCR. (**B**,**E**) IL-33 protein expression was analyzed by Western blotting with an anti-IL-33 antibody. The data are representative of experiments repeated three times with similar results. (**C**,**F**) mRNA of CCL26 in NHEKs treated with IL-4 (10 ng/mL) in the presence or absence of either tofacitinib (100, 300, and 500 nM) or JTE-052 (100, 500, and 1000 nM) for 24 h was analyzed by qRT-PCR. (**A**,**C**,**D**,**F**) Data are expressed as mean ± S.E.M.; n = 3 for each group. (**A**,**D**) Statistically significant differences between the expression of control and treated NHEKs are presented: * *p* < 0.05 (**A**,**D**). * *p* < 0.05 (**C**,**F**).

**Figure 3 jcm-09-00891-f003:**
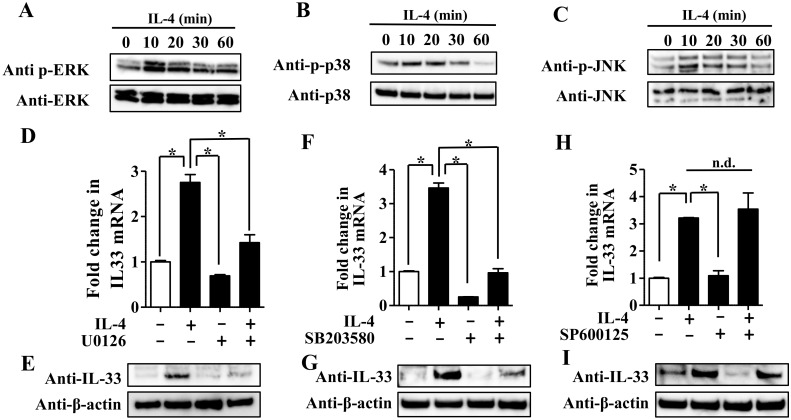
(**A**–**C**) NHEKs were treated with IL-4 (10 ng/mL) for the indicated period. Total cell lysates were prepared and subjected to Western blotting analysis with an anti-phosphorylated extracellular signal-regulated kinase (ERK)-1/2 and anti-ERK-1/2 antibody (**A**), anti-phosphorylated p-38 and anti-p38 antibody (**B**), or anti-phosphorylated JNK antibody and anti-JNK antibody (**C**). (**D**,**E**) NHEKs were treated with IL-4 (10 ng/mL) in the presence or absence of either U0126 (**D**,**E**), SB203580 (**F**,**G**), or SP600125 (**H**,**I**) for 24 h. (**D**,**F**,**H**) IL-33 expression was analyzed by qRT-PCR. Data are expressed as mean ± S.E.M.; n = 3 for each group. * *p* < 0.05. (**E**,**G**,**I**) IL-33 expression was analyzed by Western blotting with an anti-IL-33 antibody. The data are representative of experiments repeated three times with similar results.

**Figure 4 jcm-09-00891-f004:**
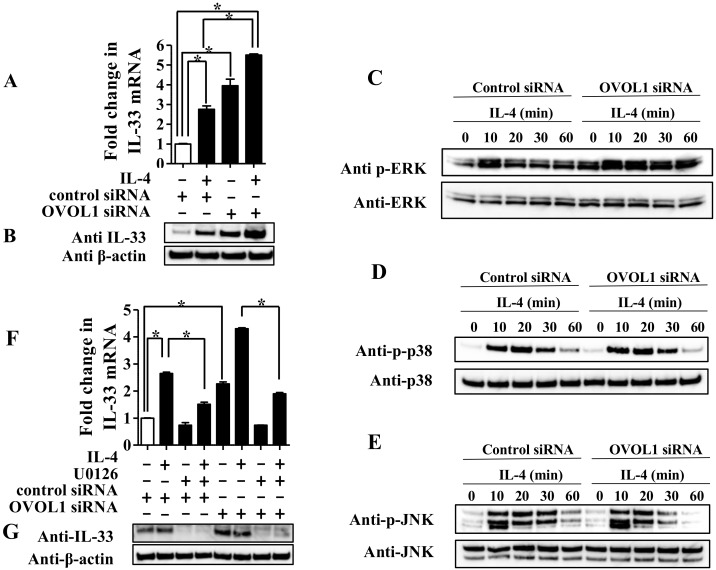
(**A**,**B**) NHEKs were transfected with control siRNA (si-control) and siRNA against OVOL1 (si-OVOL1) and subsequently treated with IL-4 (10 ng/mL) for 24 h. (**A**) IL-33 expression was analyzed by qRT-PCR. (**B**) IL-33 expression was analyzed by Western blotting with an anti-IL-33 antibody. NHEKs transfected with either si-control or si-OVOL1 were treated with IL-4 (10 ng/mL) for the indicated period and subjected to Western blotting analysis with an anti-phosphorylated ERK-1/2 and anti-ERK-1/2 antibody (**C**), anti-phosphorylated p-38 and anti-p38 antibody (**D**), or anti-phosphorylated JNK antibody and anti-JNK antibody (**E**). (**D**,**E**) NHEKs transfected with either si-control or si-OVOL1 were treated with IL-4 (10 ng/mL) for 24 h in the presence or absence of U0126. (**F**) IL-33 expression was analyzed by qRT-PCR. (**G**) IL-33 expression was analyzed by Western blotting with an anti-IL-33 antibody. (**A**,**F**) Data are expressed as mean ± S.E.M.; n = 3 for each group; * *p* < 0.05. (**C**–**E**,**G**) The data are representative of experiments repeated three times with similar results.

**Figure 5 jcm-09-00891-f005:**
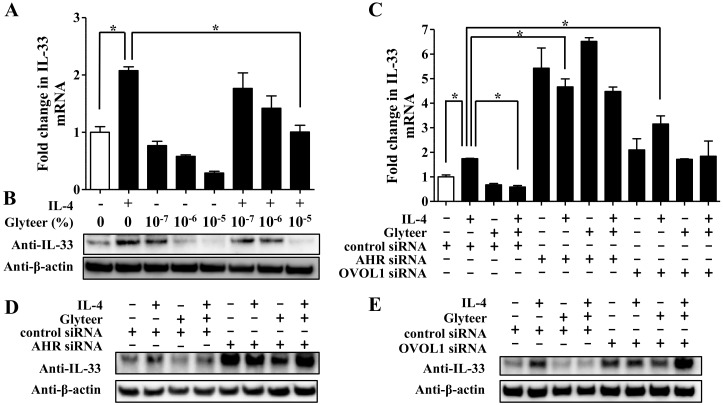
(**A**,**B**) NHEKs were treated with IL-4 (10 ng/mL) for 24 h in the presence or absence of Glyteer (10^−7^%, 10^−6^%, and 10^−5^%). (**C,D**) NHEKs transfected with si-control, siRNA against AHR (si-AHR), or si-OVOL1 were treated with IL-4 (10 ng/mL) for 24 h in the presence or absence of Glyteer (10^−5^%). (**A**,**C**) IL-33 expression was analyzed by qRT-PCR. Data are expressed as mean ± S.E.M.; n = 3 for each group; * *p* < 0.05. (**B**,**D**,**E**) IL-33 expression was analyzed by Western blotting with an anti-IL-33 antibody. The data are representative of experiments repeated three times with similar results.

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
