# Peer review of "Aryl Hydrocarbon Receptor Activation Downregulates IL-33 Expression in Keratinocytes via Ovo-Like 1"

_jcm, 2020, doi:10.3390/jcm9030891_

Round 1

Reviewer 1 Report

In this manuscript, Tsuji et al. report an interesting study about the role of AHR and OVOL1 inhibiting IL-4-induced IL33 expression and the implication for the treatment of atopic dermatitis. The topic is interesting and the paper is well written. The results are very convincing. There are some points listed below to be corrected or discussed.

Line 177, the sentence “…for 6 h (Figure 1G) …” must be “…for 8 h (Figure 1G) …” in accordance with the data from Figure 1A.

Line 222, Figure 3, the treatment of NHEKs cells with IL-4 (10 ng/ml) is for 10, 20, 30 and 60 min, according to the data showed in Figure 3A, B, C.

Lines 225-235, Figure 3D, F, the incubation time with the inhibitors of MEK and p38 is 24 h, when the activation of signaling pathways of ERK1/2 and p38 is already finished at 30 min (Figure 3A, B). It has been measured the effect of inhibitors at shorter times?

Methods.

Line 83, What means “%” in the final concentration of Glyteer ?

Line 96, What really means the sentence “NHEKs were provided from different donors and utilized in different experiments” ?

Minor points.

Line 23:  “Glyteer, an activator of activation of …” is redundant.

All along the paper some numbers of the bibliographic references are not correctly written: for example,

line 61, “…model11” instead of “…model [11].”

line 198, “19-21” instead of “[19-21]”

line 202, “22-23”

line 221, “25”

line 249, “11-14”

line 251, “15”

line 380, “16,17”

Author Response

#1 reviewer

In this manuscript, Tsuji et al. report an interesting study about the role of AHR and OVOL1 inhibiting IL-4-induced IL33 expression and the implication for the treatment of atopic dermatitis. The topic is interesting and the paper is well written. The results are very convincing. There are some points listed below to be corrected or discussed.

Line 177, the sentence “…for 6 h (Figure 1G) …” must be “…for 8 h (Figure 1G) …” in accordance with the data from Figure 1A.

> Thank you very much for this important comment. We amended the part as below:

(Line 175)

Following treatment with IL-4 (10 ng/ml) for 8 h (Figure 1G) and 24 h (Figure 1H), enhanced IL-33 expression was observed in the nucleus.

Line 222, Figure 3, the treatment of NHEKs cells with IL-4 (10 ng/ml) is for 10, 20, 30 and 60 min, according to the data showed in Figure 3A, B, C.

> Thank you very much for this important comment. We amended the part as below:

(Line 223)

NHEKs were treated with PBS (control) or IL-4 (10 ng/ml) for 10, 20, 30, and 60 min for Western blotting of phosphorylated ERK-1/2, p38, and JNK.

Lines 225-235, Figure 3D, F, the incubation time with the inhibitors of MEK and p38 is 24 h, when the activation of signaling pathways of ERK1/2 and p38 is already finished at 30 min (Figure 3A, B). It has been measured the effect of inhibitors at shorter times?

> Thank you very much for this important comment. We performed an additional experiment to examine whether the inhibitors will exhibit inhibitory effects on IL-4-induced IL-33 upregulation at shorter times. We pretreated NHEKs with U0126, SB203580, and SP600125 for 1 h and subsequently stimulated them with IL-4 for 1 h. We replaced the culture medium and then collected mRNAs after 24 h. Unfortunately, we found that the shorter incubation of the inhibitors could not inhibit IL-4-induced upregulation of IL-33 expression (data not shown).  We believe that continuous inhibition of ERK and p38 phosphorylation may be important in the mechanism.

Methods.

Line 83, What means “%” in the final concentration of Glyteer ?

> Thank you very much for this important comment. The % indicates the volume % of Glyteer, which means the volume of Glyteer / total volume of the culture medium) × 100. We amended the part as below to reflect this:

(Line 82)

Glyteer was administered to culture medium directly to reach final concentrations of 10−5%, 10−6%, and 10−7%. The % here indicates the volume % of Glyteer in the culture medium.

Line 96, What really means the sentence “NHEKs were provided from different donors and utilized in different experiments” ?

> Thank you very much for this comment. We wanted to indicate that pooled NHEKs were not utilized in this study. We deleted the sentence to avoid confusion.

Minor points.

Line 23:  “Glyteer, an activator of activation of …” is redundant.

> Thank you very much for this comment. We amended the sentence as below:

(Line 23)

Glyteer, an activator of aryl hydrocarbon receptor (AHR)

All along the paper some numbers of the bibliographic references are not correctly written: for example,

> Thank you very much for this comment. We amended the corresponding parts.

line 61, “…model11” instead of “…model [11].”

line 198, “19-21” instead of “[19-21]”

line 202, “22-23”

line 221, “25”

line 249, “11-14”

line 251, “15”

line 380, “16,17”

We thank you again for your constructive comments.

Best regards,

Gaku Tsuji

Research and Clinical Center for Yusho and Dioxin,

Kyushu University Hospital,

Department of Dermatology,

Graduate School of Medical Sciences,

Kyushu University

Reviewer 2 Report

This is an interesting study addressing the functional relevance of aryl hydrocarbon receptor (AHR) signaling for the IL-4 induced modulation of IL-33 expression in human epidermal keratinocytes. The paper is well written and the experimental data are presented in an appropriate manner. However, not all of the presented data are novel observations and the major finding, i.e. AHR downregulates IL-33 expression, is in stark contrast to published literature (not even mentioned by the authors). 

Major points of concern:

Results section, first sentence “To investigate the regulatory mechanism of IL-33 expression in human keratinocytes under diseased conditions that recapitulate AD, we examined this expression in NHEKs treated with IL-4.” Please attenuate this statement. Treatment of one cutaneous cell population with just one cytokine is fore sure not suitable to mimic processes taking place in such a complex inflammatory disease like atopic dermatitis, which involves multiple cutaneous cell populations incl. immune cells, as well as dozens of lipid mediators, cytokines, etc.

The finding that IL-4 treatment induces the expression of IL-33 in NHEKs is already published (Meephansan et al. 2012 J Invest Dermatol 132 , Du et al. 2016 J Interfer Cytok Res 36) and thus not new.

Several publications exists reporting that AHR activation induces IL-33 expression in different cells and tissues, including the epidermis, probably by binding to functional dioxin-responsive element in the gene promoter (Hidaka et al. 2017 Nature Immunol; Ishihara et al. 2019 Toxicol Sci; Weng et al. 2018 Allergy). How does this fit to the glyteer data shown in figure 5? Since glyteer contains hundreds of compounds and mixture effects cannot be excluded, the authors should repeat these experiments with an established AHR agonist, such as TCDD, FICZ etc.

Used concentration of JNK inhibitor vs. concentration of p38 and MEK inhibitor:

Even though, the IC50 value of all three inhibitors (U0126, SB203580, SP600125) for their primary target kinase is clearly below 0.1 µM, the authors treated the cells with 10 µM of the p38 and MEK inhibitors, but with only 1 µM of the JNK inhibitor (which had no effect in the respective experiments; fig. 3). Why?

Figure 4: Do not understand the relevance of these experiments with regards to the signaling pathways that are activated by either IL-4 or AHR agonists in normal human keratinocytes or skin. The role of the OVOL1 transcription factor in this process is not getting clear. How does OVOL1 interfere with ERK phosphorylation or protein kinase activity in general?

Author Response

#2 reviewer

This is an interesting study addressing the functional relevance of aryl hydrocarbon receptor (AHR) signaling for the IL-4 induced modulation of IL-33 expression in human epidermal keratinocytes. The paper is well written and the experimental data are presented in an appropriate manner. However, not all of the presented data are novel observations and the major finding, i.e. AHR downregulates IL-33 expression, is in stark contrast to published literature (not even mentioned by the authors).

Major points of concern:

Results section, first sentence “To investigate the regulatory mechanism of IL-33 expression in human keratinocytes under diseased conditions that recapitulate AD, we examined this expression in NHEKs treated with IL-4.” Please attenuate this statement. Treatment of one cutaneous cell population with just one cytokine is fore sure not suitable to mimic processes taking place in such a complex inflammatory disease like atopic dermatitis, which involves multiple cutaneous cell populations incl. immune cells, as well as dozens of lipid mediators, cytokines, etc.

> Thank you very much for this important comment. We amended the part as below:

(Lines 167–168)

To investigate the regulatory mechanism of IL-33 expression induced by IL-4, a key cytokine in the pathogenesis of AD [1], we examined this expression in NHEKs treated with IL-4.

The finding that IL-4 treatment induces the expression of IL-33 in NHEKs is already published (Meephansan et al. 2012 J Invest Dermatol 132 , Du et al. 2016 J Interfer Cytok Res 36) and thus not new.

 > Thank you very much for this important comment. We cited the reports and amended the part as below:

(Lines 177–179)

These findings are consistent with previous reports regarding IL-4-induced upregulation of IL-33 expression in NHEKs [18,19].

(Lines 531–535)

  1. Meephansan J, Tsuda H, Komine M et al. Regulation of IL-33 expression by IFN-γ and tumor necrosis factor-α in normal human epidermal keratinocytes. J Invest Dermatol 2012, 132, 2593-2600.

  1. Du HY, Fu HY, Li DN, et al. The expression and regulation of interleukin-33 in human epidermal keratinocytes: A new mediator of atopic dermatitis and its possible signaling pathway. J Interferon Cytokine Res 2016, 36, 552-562.

Several publications exist reporting that AHR activation induces IL-33 expression in different cells and tissues, including the epidermis, probably by binding to functional dioxin-responsive element in the gene promoter (Hidaka et al. 2017 Nature Immunol; Ishihara et al. 2019 Toxicol Sci; Weng et al. 2018 Allergy). How does this fit to the glyteer data shown in figure 5?

> Thank you very much for this important comment. As you pointed out, it has been shown that ligand-activated AHR mediates the induction of IL-33 via a DRE located in the IL-33 promoter region. In addition, TCDD, diesel exhaust particles (DEPs), and benzo(a)pyrene (BaP) induced the upregulation of IL-33 expression (Hidaka et al. 2017 Nature Immunol; Ishihara et al. 2019 Toxicol Sci; Weng et al. 2018 Allergy). Recent evidences suggest AHR ligands trigger differential responses leading to cell-type specific AHR-dependent effects (Sonia Mulero-Navarro and Pedro M. Fernandez-Salguero, New Trends in Aryl Hydrocarbon Receptor Biology

Front. Cell Dev. Biol., 2016). Ishihara et al. utilized murine macrophage and Weng et al. utilized primary bronchial epithelial cells. Therefore, there is a possibility that a response of IL-33 expression induced by AHR activation in keratinocytes may differ from that in macrophages and primary bronchial epithelial cells. Hidaka et al. have reported that BaP upregulate IL-33 expression in murine keratinocytes, but they have also reported that treatment with 6-formylindolo[3,2-b]carbazole (FICZ), an endogenous AHR ligand, does not upregulate IL-33 expression but instead downregulates it in differentiated human keratinocytes compared with that in undifferentiated human keratinocytes. Therefore, we agree with your comment that whether AHR activation results in the inhibition of IL-33 expression is still controversial.

It has been thought that the type of AHR ligand is an important factor determining the response of AHR signaling (Furue M and Tsuji G, Int J Mol Sci, 2019). TCDD, DEPs, and BaP activate AHR and upregulate CYP1A1 expression. CYP1A1 attempts to metabolize them, but its continuous efforts are unsuccessful because they are structurally stable. The metabolism by CYP1A1 generates excessive amounts of reactive oxygen species (ROS) and subsequently induces oxidative damage in the cell. To survive during oxidative stress, antioxidative machinery is simultaneously activated after AHR activation in the cells. Ligation of AHR also activates antioxidative transcription factor NRF2 and upregulates the expression of antioxidative enzymes. TCDD, DEPs, and BaP activate the AHR-NRF2 battery, but the powerful AHR-CYP1A1-ROS pathway may induce far more oxidative stress that cannot be extinguished by the AHR-NRF2 antioxidative system. In contrast, many phytochemical AHR ligands stimulate the AHR-NRF2 battery more strongly than the AHR-CYP1A1-ROS pathway and exert antioxidative effects. Therefore, we believe that oxidative stress induced by AHR activation has a key role in the mechanism of AHR-mediated IL-33 expression. Also, considering that oxidative stress serves as a key checkpoint for IL-33 production, which is inhibited by NRF2 activation (Uchida M et al., Allergy 2017), there is a possibility that AHR-NRF2 activation by Glyteer may also be involved in the inhibitory effect on the upregulation of IL-33 expression. Taking these findings together, we amended part of the discussion as below:

(Lines 390–418)

Therefore, AHR is likely to mediate IL-33 expression transcriptionally in NHEKs; however, further investigations will be needed to demonstrate this.

It has been shown that the induction of IL-33 is mediated via AHR-binding sites located in the IL-33 promoter region. Furthermore, 2,3,7,8-tetrachlorodibenzodioxin (TCDD), diesel exhaust particles (DEPs), and benzo(a)pyrene (BaP) have been shown to induce the upregulation of IL-33 expression [40–42]. In contrast, it has been reported that treatment with 6-formylindolo[3,2-b]carbazole (FICZ), an endogenous AHR ligand, does not upregulate IL-33 expression but instead downregulates it in differentiated human keratinocytes compared with the case in undifferentiated human keratinocytes [38]. Therefore, whether AHR activation results in the inhibition of IL-33 expression is still controversial; however, it has been shown that the type of AHR ligand is an important factor determining the response of AHR signaling [43]. We also examined whether tapinarof, a potent AHR activator [44] that is utilized clinically in the treatment of psoriasis and atopic dermatitis [45,46], has the same effect as Glyteer on IL-4-induced upregulation of IL-33 expression. As shown in Supplementary Figure 7, tapinarof inhibited the IL-4-induced upregulation of IL-33 expression in NHEKs.

It has been shown that the response of AHR signaling is modified by oxidative stress [41]. TCDD, DEPs, and BaP generate excessive amounts of reactive oxygen species (ROS) and subsequently induce oxidative damage in the cell [47–49], since their metabolism by CYP1A1 is unsuccessful because they are structurally stable. Ligation of AHR also activates nuclear factor erythroid 2-related factor 2 (NRF2) and upregulates the expression of antioxidative enzymes [41]. TCDD, DEPs, and BaP activate the AHR-NRF2 battery, but the AHR-CYP1A1 activation may cause far more oxidative stress that cannot be extinguished by the AHR-NRF2 battery. In contrast, many phytochemical AHR ligands stimulate the AHR-NRF2 battery more strongly than the AHR-CYP1A1-mediated ROS production and exert antioxidative effects [44]. Therefore, we believe that oxidative stress induced by AHR activation has an important role in the mechanism of AHR-mediated IL-33 expression. Considering that oxidative stress serves as a key checkpoint for IL-33 production, which is inhibited by NRF2 activation [50], there is a possibility that AHR-NRF2 activation by Glyteer may also be involved in the inhibitory effect on the upregulation of IL-33 expression.

Since glyteer contains hundreds of compounds and mixture effects cannot be excluded, the authors should repeat these experiments with an established AHR agonist, such as TCDD, FICZ etc.

> Thank you very much for this important comment. As you suggested, we performed an additional experiment using tapinarof, a potent AHR activator that is utilized clinically in the treatment of psoriasis and atopic dermatitis. As shown in Supplementary Figure 7, tapinarof also inhibited IL-4-induced upregulation of IL-33 expression in NHEKs. Based on this comment, we amended the part as below:

(Lines 402–405)

We also examined whether tapinarof, a potent AHR activator that is utilized clinically in the treatment of psoriasis and atopic dermatitis [45,46], has the same effect as Glyteer on IL-4-induced upregulation of IL-33 expression. As shown in Supplementary Figure 7, tapinarof also inhibited IL-4-induced upregulation of IL-33 expression in NHEKs.

Used concentration of JNK inhibitor vs. concentration of p38 and MEK inhibitor:

Even though, the IC50 value of all three inhibitors (U0126, SB203580, SP600125) for their primary target kinase is clearly below 0.1 µM, the authors treated the cells with 10 µM of the p38 and MEK inhibitors, but with only 1 µM of the JNK inhibitor (which had no effect in the respective experiments; fig. 3). Why?

> Thank you very much for this important comment. Since we found that 10 µM SP600125 affected cell viability (Supplementary Figure 1), we utilized 1 µM JNK inhibitor in this experiment. In accordance with a previous paper (ERK1/2 regulates epidermal chemokine expression and skin inflammation. Pastore S, Mascia F, Mariotti F, Dattilo C, Mariani V, Girolomoni G. J Immunol. 2005), the researchers also utilized 1 µM SP600125 and 10 µM SB203580 for NHEKs. In addition, 10 µM U0126 was also utilized for experiments using NHEKs (Isolation and functional analysis of a keratinocyte-derived, ligand-regulated nuclear receptor comodulator. Flores AM, Li L, Aneskievich BJ. J Invest Dermatol. 2004). Therefore, we think that the concentration of the inhibitors is appropriate for the experiments.

Figure 4: Do not understand the relevance of these experiments with regards to the signaling pathways that are activated by either IL-4 or AHR agonists in normal human keratinocytes or skin. The role of the OVOL1 transcription factor in this process is not getting clear. How does OVOL1 interfere with ERK phosphorylation or protein kinase activity in general?

> Thank you very much for this important comment. As far as we know, there is no report regarding the mechanism by which OVOL1 interferes with ERK phosphorylation. Since OVOL1 is a transcription factor, we assumed that it may negatively regulate the factors inhibiting ERK phosphorylation, such as MAPK scaffolding proteins; however, further study will be needed to clarify this. Based on this comment, we amended the part of the discussion as below:

(Lines 374–376)

 Since OVOL1 is a transcription factor, we assumed that it may negatively regulate factors inhibiting ERK phosphorylation such as MAPK scaffolding proteins [37]; however, further study will be needed to confirm this.

We thank you again for your constructive comments.

Best regards,

Gaku Tsuji

Research and Clinical Center for Yusho and Dioxin,

Kyushu University Hospital,

Department of Dermatology,

Graduate School of Medical Sciences,

Kyushu University

Round 2

Reviewer 2 Report

the authors have adequately addressed all my points of concern. Therefore, I recommend the publication of this work in its present form.